# A novel approach for modelling and classifying sit-to-stand kinematics using inertial sensors

**Maitreyee Wairagkar**[1,2,3], **Emma Villeneuve**[4], **Rachel King**[3], **Balazs Janko**[3], **Malcolm Burnett**[5], **Veena Agarwal**[5], **Dorit Kunkel**[5], **Ann Ashburn**[5], **R. Simon Sherratt**[3], **William Holderbaum**[6], **William S. Harwin**[3]*

**1** Department of Mechanical Engineering, Imperial College London, London, United Kingdom, **2** Care Research and Technology Centre, UK Dementia Research Institute, London, United Kingdom, **3** Department of Biomedical Engineering, University of Reading, Reading, United Kingdom, **4** Univ. Grenoble Alpes, CEA, LETI, DTBS, LS2P, Grenoble, France, **5** School of Health Sciences, University of Southampton, Southampton, United Kingdom, **6** Faculty of Science and Engineering, Manchester Metropolitan University, Manchester, United Kingdom

* w.s.harwin@reading.ac.uk

**Data Availability Statement:** The code used in this study is available on GitHub at: https://github.com/maitreyeew/Modelling-movement-kinematics-using-inertial-sensors.

## Abstract

Sit-to-stand transitions are an important part of activities of daily living and play a key role in functional mobility in humans. The sit-to-stand movement is often affected in older adults due to frailty and in patients with motor impairments such as Parkinson's disease leading to falls. Studying kinematics of sit-to-stand transitions can provide insight in assessment, monitoring and developing rehabilitation strategies for the affected populations. We propose a three-segment body model for estimating sit-to-stand kinematics using only two wearable inertial sensors, placed on the shank and back. Reducing the number of sensors to two instead of one per body segment facilitates monitoring and classifying movements over extended periods, making it more comfortable to wear while reducing the power requirements of sensors. We applied this model on 10 younger healthy adults (YH), 12 older healthy adults (OH) and 12 people with Parkinson's disease (PwP). We have achieved this by incorporating unique sit-to-stand classification technique using unsupervised learning in the model based reconstruction of angular kinematics using extended Kalman filter. Our proposed model showed that it was possible to successfully estimate thigh kinematics despite not measuring the thigh motion with inertial sensor. We classified sit-to-stand transitions, sitting and standing states with the accuracies of 98.67%, 94.20% and 91.41% for YH, OH and PwP respectively. We have proposed a novel integrated approach of modelling and classification for estimating the body kinematics during sit-to-stand motion and successfully applied it on YH, OH and PwP groups.

## Introduction

Kinematic modelling of human body motion gives an insight into specific movements which can be used for studying human gait and posture, assessing the quality of movements, for

**Funding:** This work was funded by the UK Engineering and Physical Sciences Research Council (EPSRC) through SPHERE IRC under Grant EP/K031910/1 (https://irc-sphere.ac.uk/) awarded to the Universities of Bristol, Reading and Southampton. The funders had no role in study design, data collection and analysis, decision to publish, or preparation of the manuscript.

**Competing interests:** The authors have declared that no competing interests exist.

monitoring and diagnostic purposes and developing rehabilitation strategies. The reviews by Yang *et al.* (2010) [1] and Fong *et al.* (2010) [2] suggest that several studies have shown some initial results for monitoring and rehabilitation of people with motor functional impairments by examining different categories of motions including static postures such as sitting, standing, lying down; cyclic dynamic activities such as walking, running, stairs climbing; as well as transitions such as sit-to-stand and stand-to-sit to move between static and dynamic activities. Out of these motions, investigating the kinematics of sit-to-stand transitions is important because of their significance in functional mobility [3]. The sit-to-stand transitions have been studied widely in children [4], adults [5] and older adults [6, 7] for assessing their mobility in activities of daily living [8]. The sit-to-stand transitions are an important part to activities of daily living with an estimated frequency of $60 \pm 22$ for healthy adults [9]. Studying these transitions is also beneficial for clinical monitoring of patients with motor disorders such as Parkinson's disease [10–12], predicting falls [13–15] and frailty [16, 17] in older adults. Hence, there is an increasing research interest in investigating the biomechanics and kinematics of these postural transitions. Sit-to-stand is a good representative transition that is easy to record in a controlled environment and is also primarily a planar transition, hence in this study, we model sit-to-stand kinematics and classify these transitions.

The kinematics of human motions can be estimated using inertial sensors, such as accelerometers and gyroscopes that provide a reliable, cost effective and wearable alternative to motion capture systems for detecting posture and movements [18–20]. Inertial sensors have been used to identify sit-to-stand transitions [5, 21] and also to extract biomechanical information [22]. Often, parameters such as transition duration, angular and linear velocities, trunk tilt range, spectral edge frequencies and entropy values are used to evaluate functional performance of sit-to-stand and stand-to-sit transitions [23].

The sit-to-stand transitions can be identified by using a single or multiple inertial sensors positioned on various locations such as the waist, hip or lower back [5, 24, 25] and chest [26, 27]. Various classification schemes such as the wavelet methods [24, 28] and Support Vector Machines (SVM) [12] have been used to identify sit-to-stand and stand-to-sit from the inertial sensors.

Most of the previous studies focus on classification of the sit-to-stand transitions and very few model their kinematics. A theoretical model of sit-to-stand has been proposed by Musić *et al.* (2008) [22]. Assessing a movement by modelling its kinematics is important for diagnosing and determining the severity of motor impairment, devising rehabilitation strategy, and monitoring patient's progress and outcomes of the intervention [29]. The activity classification on the other hand, enables recognition of different movements [30] which is useful in developing assistive technologies. Combining the modelling and classification can help in pinpointing the problem areas in the movement as well as assessing the change in the motion kinematics in the affected population. However, to our knowledge, there are no methods where modelling of kinematics and classification of sit-to-stand transitions are explored via inter-dependent algorithms. The sit-to-stand kinematics are typically modelled by placing one sensor per segment [22, 31–33]; additionally, multiple force sensors are also used [22]. In this study, we show that the body kinematics can be modelled using only two wearable inertial sensors, instead of placing sensors on all the segments of the body or more traditional five sensor configuration with sensors on two legs, two hands and waist [14]. To our knowledge, this has not been evaluated on lower limb activities.

In our previous work, we presented a two-segment model to estimate kinematics of upper limb with an inertial sensor on each segment [34]. In this study, we expand upon this two-segment upper limb model [34] to develop a three-segment model for estimating sit-to-stand transition kinematics by including a classification based modelling approach using only two

inertial sensors. Unlike our previous work, in this study, we not only integrate classification of sit-to-stand transitions in the modelling process, but also use fewer number of sensors than the body segments being modelled. Additionally, we also demonstrate the generalisability of our novel integrated kinematics estimation approach using three different participant groups with varied ages and motor abilities including young healthy adults, older healthy adults and people with Parkinson's disease.

The aims of this work are:

1. To design an integrated approach for monitoring and classification of sit-to-stand transitions and validate this model by comparing the results with motion capture reference data.

2. To apply this method to estimate the three-segment sit-to-stand angular kinematics of older healthy participants and people with Parkinson's using only two inertial sensors. This is appropriate for people with motor-related physiological conditions to better understand their condition and the affected motion kinematics.

We have chosen these groups to represent the problems that people tend to develop later in the life. We demonstrate this using a novel method of modelling human motion kinematics using only two inertial sensors with triaxial accelerometer and triaxial gyroscope so as to understand sit-to-stand movement in healthy individuals and individuals with motor impairments. We apply this model to accurately estimate the angular kinematics and classify sit-to-stand and stand-to-sit movements with three-segment body model consisting of the shank, thigh and back. We have reduced the number of sensors to make the system more comfortable to wear and facilitate measurements for longer duration, while reducing the energy requirements and sensor setup time.

## Methods

### Parametric modelling and estimation of angular kinematics

The sit-to-stand transition angular kinematics for the three segments: the shank, thigh and back were modelled using the measurements from two inertial sensors placed on the shank and back. This was achieved in two stages: 1) modelling the relationship between the limb kinematics and sensor measurements; 2) parameter estimation using the model. The parameter estimation was done in further two stages. First, the shank and back kinematics were estimated directly from the inertial measurements. Second, as there was no sensor on the thigh, the corresponding kinematics were reconstructed using the previous outcome. A combined classification based approach was used to estimate the thigh kinematics.

**Estimation of the angular kinematics for the shank and the back.** *Kinematic model for the shank and the back*. We estimated the kinematics during sit-to-stand using a 2-dimensional three segment model of a body in the sagittal plane. We have chosen a 2-dimensional model because the sit-to-stand motion occurs mainly in the sagittal plane and hence contains the maximum information about the motion. Most of the studies in the literature investing the kinematics of the three body segments also assumed that the movement was restricted to the sagittal plane [2]. This 2-dimensional model is sufficient for our purposes to study the kinematics of sit-to-stand transitions and classify them in the all the three participant groups. The third dimension might not provide additional information about sit-to-stand especially in the less dynamic older healthy and people with Parkinson's groups. This is discussed further in the Discussion section. The first, second and third segment represent the shank (S), thigh (T) and back (B) respectively as shown in the Fig 1. Two inertial sensors with a triaxial accelerometer and a triaxial gyroscope each were placed on the shank and back. The inertial sensors were

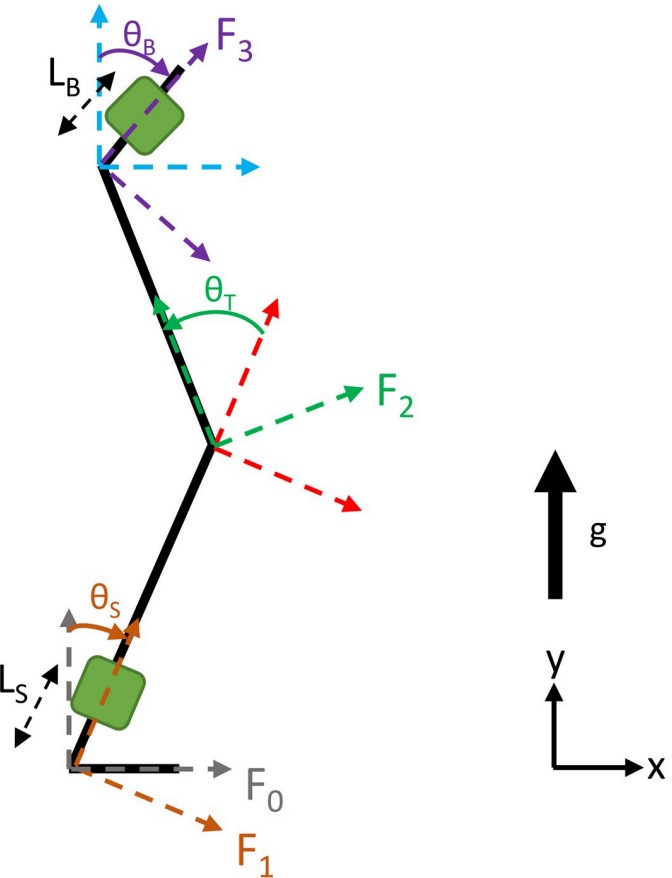

**Fig 1. Leg and trunk three-segment 2-dimensional model in the sagittal plane.** $\theta_S$ is the angle for the shank, $\theta_T$ is the angle for the thigh and $\theta_B$ is the angle for the back. Green squares on the shank and back segment represent the inertial sensors. $L_S$ and $L_B$ denote the distance of the sensor placement on the shank from the ankle and the back from the hip respectively.

placed at a distance $L_S$ from the ankle on the shank and at a distance $L_B$ from the hip on the back. Angular kinematics including position $\theta_i$, velocity $\omega_i$ and acceleration $\alpha_i$, $i \in \{S, T, B\}$ for each of the three segments were to be determined. The $\theta_i$, $\omega_i$ and $\alpha_i$ are functions of time where $\theta_S$, $\theta_T \in [0, \pi/2]$ and $\theta_B \in [-\pi/2, \pi/2]$.

For estimation of the shank and back angular kinematics, the translation of the ankle and hip joints is neglected and the same model is applied to both the segments. Such approximation is straightforward for the ankle as the foot is on the ground during sit-to-stand transitions and the shank pivots around the ankle; it is also reasonable for the hip and necessary for the kinematic estimation from only one sensor. Vectors are written in bold; the superscript of the vector refers to the coordinate frame in which it is written, for e.g. $^0\mathbf{g}$; if omitted, the vector is written in the reference frame {0}; matrices are written in capital letters. Rotation matrices from frame $j$ to frame $k$ are written as $^jR_k$.

The sensor is located at $^1\mathbf{d}_i = [0, L_i, 0] \in \mathbb{R}^3$, $i \in \{S, B\}$ in the local frame, and $^0\mathbf{d}_i = {}^0R_1{}^1\mathbf{d}_i$ in the reference frame. The linear acceleration $\mathbf{a}_i \in \mathbb{R}^3$, $i \in \{S, B\}$ of the sensor, written in the reference frame is given by Eq (1) [35], where $\boldsymbol{\omega}_i \in \mathbb{R}^3$ is the angular velocity and $\dot{\boldsymbol{\omega}}_i \in \mathbb{R}^3$ is the angular acceleration.

$$^0\mathbf{a}_i = {}^0\dot{\boldsymbol{\omega}}_i \times {}^0\mathbf{d}_i + {}^0\boldsymbol{\omega}_i \times ({}^0\boldsymbol{\omega}_i \times {}^0\mathbf{d}_i) \tag{1}$$

Hence, the linear accelerations measured by accelerometer on the shank and back are modelled by Eq (2)

$$^1\mathbf{a}_i = {^1R_0}({^0\mathbf{a}_i} + {^0\mathbf{g}})$$ (2)

where, $^1R_0 = \begin{bmatrix} \cos(\theta_s) & \sin(\theta_s) & 0 \\ -\sin(\theta_s) & \cos(\theta_s) & 0 \\ 0 & 0 & 1 \end{bmatrix}$ is the rotation transformation between the frame 0

(world) and frame 1 (sensor) and gravity component in the reference frame $^0\mathrm{g} = \begin{bmatrix} 0 \\ g \\ 0 \end{bmatrix}$ point-

ing upwards. Thus, for the shank and the back, we get (3) [34].

$$^1\mathbf{a}_i = \begin{pmatrix} g\sin(\theta_i) - L_i\alpha_i \\ g\cos(\theta_i) - L_i\omega_i^2 \end{pmatrix}, \quad \mathbf{a}_i \in \mathbb{R}^3, \quad i \in \{S, B\}$$ (3)

In order to estimate the angle of the back relative to the reference frame, one can apply the shank model if the acceleration of the hip is neglected.

*Extended Kalman filter to estimate the angular kinematics.* Expanding on our previous work [34] for estimating upper limb kinematics, we use extended Kalman filter (EKF) for obtaining the angle $\theta_i$, the angular velocity $\omega_i$ and the angular acceleration $\alpha_i$, $i \in \{S, B\}$ for the shank and back independently. The state vector for the EKF is given by $\mathbf{x}_t = [\theta_i, \omega_i, \alpha_i]^T$, where $\mathbf{x}_t$ is a function of time. The transition matrix $F$ that describes a link between a new state sample from the previous one is given by Eq (4) [36].

$$F = \begin{bmatrix} 1 & \Delta T & \dfrac{\Delta T^2}{2} \\ 0 & 1 & \Delta T \\ 0 & 0 & 1 \end{bmatrix}$$ (4)

where, $\Delta T$ is sampling period (in this case 0.02 s). The process model for a single link for EKF is given by Eq (5).

$$\mathbf{x}_t = F\mathbf{x}_{t-1} + \mathbf{v}_{t-1}$$ (5)

where, $\mathbf{v}_t \sim \mathcal{N}(0, Q)$ is the process noise, which is a centred Gaussian noise of a covariance matrix Q, where we have chosen

$$Q = \begin{bmatrix} (\Delta T^2)^2 & 0 & 0 \\ 0 & (0.1\Delta T)^2 & 0 \\ 0 & 0 & (0.04)^2 \end{bmatrix}.$$

We want to estimate $\theta_i$, $\omega_i$ and $\alpha_i$ from the measurements observed from one accelerometer and one gyroscope. Using the relationship between the linear acceleration $a_x$ on x-axis, $a_y$ on y-axis obtained from the accelerometer measurements and the angular kinematics given in Eq (3); and the angular velocity measurement obtained directly from the z-axis of the gyroscope

$gyr_z$, we can establish the relation given in Eq (6) for each time point $t$.

$$\underbrace{\begin{pmatrix} a_{x,i} \\ a_{y,i} \\ gyr_{z,i} \end{pmatrix}}_{z_t} = \underbrace{\begin{pmatrix} g\sin(\theta_i) - L\,\alpha_i \\ g\cos(\theta_i) - L\,\omega_i^2 \\ \omega \end{pmatrix}}_{H(x_t)} \tag{6}$$

where, $i \in \{S, B\}$.

The measurements obtained from the inertial sensors are noisy and hence the measurement model for the EKF is given by Eq (7).

$$z_t = H(x_t) + w_t \tag{7}$$

where, $w_t \sim \mathcal{N}(0, R)$ and $R = \begin{bmatrix} \left(\frac{g}{10}\right)^2 & 0 & 0 \\ 0 & \left(\frac{g}{10}\right)^2 & 0 \\ 0 & 0 & (0.005)^2 \end{bmatrix}$ is the covariance matrix of

Gaussian measurement noise resulting from the accelerometer and gyroscope. The exact values of Q and R were fine tuned manually offline, which is a common approach of determining these parameters of Kalman filters [37].

The process model given in (5) is used in the prediction step for the EKF and the measurement model given in (7) is used in the updating step of the EKF.

This EKF model is used independently for obtaining shank kinematics $\theta_S$, $\omega_S$ and $\alpha_S$ and back kinematics $\theta_B$, $\omega_B$ and $\alpha_B$ using the measurements from inertial sensors placed on the shank and the back respectively.

**Estimation of the angular kinematics for the thigh.** The thigh movement is not measured using an inertial sensor and hence, its angular kinematics could not be estimated directly. We used a classification-based approach by using the kinematics from the shank and back to identify four classes: sitting, standing, sit-to-stand and stand-to-sit. The thigh angular kinematics were estimated for each of the classes separately because we observed from the reference data that, the kinematics for each class could be modelled by a different function. A two-tiered classification scheme was used. The first classifier distinguished between a stationary state (sitting and standing) and transition state (sit-to-stand and stand-to-sit). The second classifier classified sit-to-stand and stand-to-sit movements; and based on that, a probabilistic approach was used to determine sitting or standing state. The analysis pipeline of the estimation of the kinematics of the shank, back and thigh is given in Fig 2.

*Classification 1- automatic segmentation and identification of stationary and transition states.* The first classifier segmented and identified the data with multiple sit-to-stand movements into individual stationary and transition states automatically from the shank and back angular kinematics. A one dimensional feature vector was created by first, multiplying together the angular kinematics $\theta_i$ and $\omega_i$, $i \in \{S, B\}$ for the shank and back; second, taking absolute value of the feature vector; and third, smoothing it with a moving average filter $y_t = 1/n \sum_{i=0}^{n-1} x_{t+i}$ with n = 5 to eliminate trivial peaks and avoid spurious misclassification. This feature vector had values close to zero during the stationary states and higher values during transition state. Multiplying $\theta_i$ and $\omega_i$ together to obtain this feature vector made the values in the stationary state even smaller and enhanced the difference between stationary and transition states. A threshold for classification was determined automatically from the right edge of

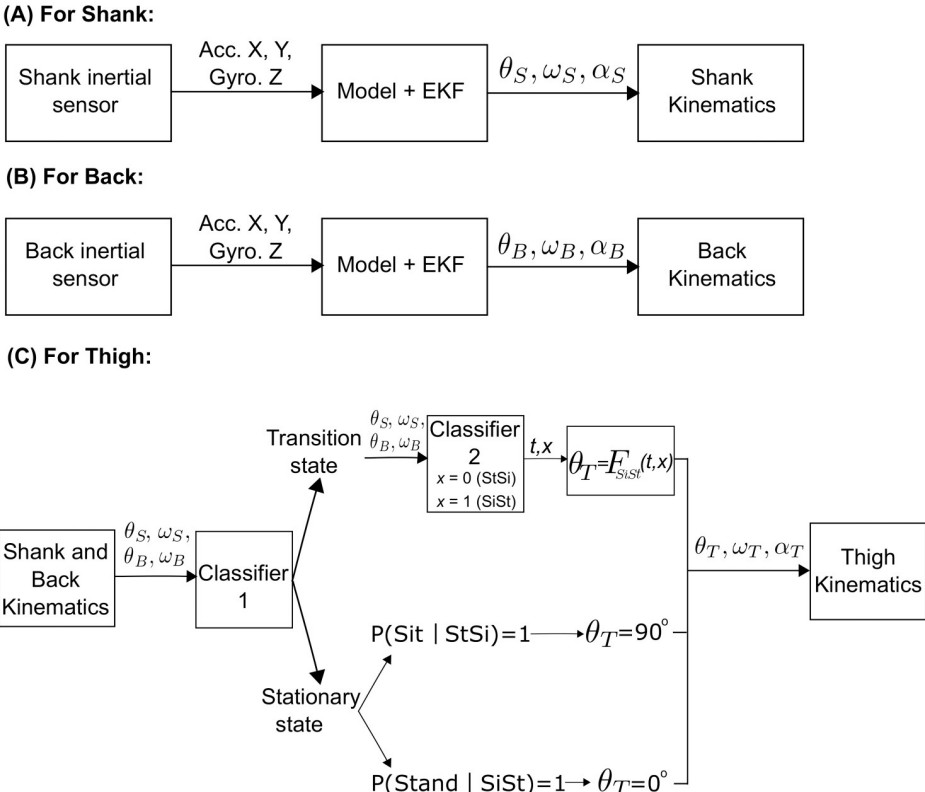

**Fig 2. Processing steps for estimation of shank, back and thigh kinematics.** (A) Shank kinematics estimation process using the model and EKF. (B) Back kinematics estimation process using the model and EKF. (C) Thigh kinematics estimation process by integrating the results of (A) and (B) and two-tiered classification scheme to segment and identify sit-to-stand (SiSt), stand-to-sit (StSi), sitting and standing states.

first bin in the histogram of the peaks from feature vector. Bin width for the histogram was obtained using Scott's rule [38]. Histogram was computed using the MATLAB (The Math-Works, Inc., Natick, Massachusetts, US) function *histogram()*. The first bin of the histogram captured feature values of stationary states which were close to zero, hence the right edge of the first bin computed using Scott's rule gave the threshold value. A threshold-based binary linear classification was performed to identify stationary state for feature values below the threshold and transition state for feature values above the threshold. Spurious misclassifications were identified and corrected automatically by finding one or two samples that had a different class from their neighbouring samples. Based on the classification results, a series of sit-to-stand movements was segmented into stationary and transition states. Fig 3 shows the two-tier classification scheme.

*Classification 2—classifying sit-to-stand, stand-to-sit, sitting and standing states using unsupervised learning.* Once the stationary and transition state segments were identified, the second classification was done on the transition segments to classify sit-to-stand and stand-to-sit states. We employed unsupervised learning using k-means clustering [39] to automatically classify sit-to-stand and stand-to-sit states. Four-dimensional features were used for k-means clustering. The four features obtained for each transition segment were as follows: the slope of the linear regression of the segment $\omega_S$ (such that it captures the amount of increase or decrease in $\omega_S$ in the selected transition segment which differs between sit-to-stand and stand-to-sit), the slope of the linear regression of the segment $\omega_B$, the difference between the start

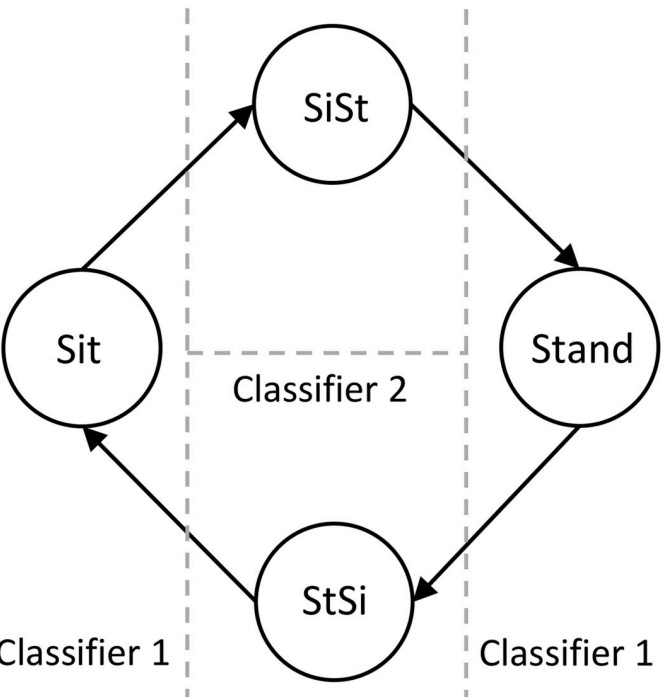

**Fig 3. Classification scheme for sit, stand, sit-to-stand and stand-to-sit.** State diagram representing the classification scheme for sit, stand, sit-to-stand and stand-to-sit. The vertical dashed line represents the threshold for the Classifier 1 to classify the stationary state and the transition state. The horizontal dashed line represents the classification boundary for Classifier 2 to classify the transition state into sit-to-stand and stand-to-sit.

and end points of the segment $\theta_S$ and the difference between the start and end points of the segment $\theta_B$. After classification, the labels were assigned to the two clusters post-hoc based of their values. The classification was performed on individual participant independently with limited number of trials. The advantage of using unsupervised learning was that only single trial from an individual participant with as few as two sit-to-stand transitions could be used to classify both the states correctly. The sit-to-stand and stand-to-sit states in two participants with Parkinson's who managed to perform only two sit-to-stand transitions were also classified correctly using unsupervised learning. As opposed to this, supervised learning requires several examples of sit-to-stand transitions to train the classifier.

Based on whether the previous transition segment was sit-to-stand or stand-to-sit, the stationary state segment was classified into sitting or standing state as follows:

- If the previous transition segment was sit-to-stand, then the probability of standing state in the current stationary segment was set to 1 and hence, the segment was classified as standing state.

- If the previous transition segment was stand-to-sit, then the probability of the sitting state in the current stationary segment was set to 1 and hence, the segment was classified as sitting state.

We based the probability of identifying stationary states on the class of the previous transition state because the angular velocity and angular acceleration both are zero during sitting and standing and the angle of shank and back varies according to individual's posture. Fig 2C shows the classification scheme used for estimating the thigh kinematics.

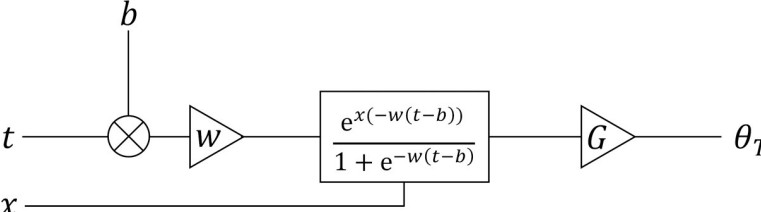

**Fig 4. Model to estimate thigh angle.** A model based on an artificial neuron with a sigmoid activation function to estimate stand-to-sit thigh angle where, $t$ is the input time segment of stand-to-sit transition, weight $w$ determines the speed of transition, bias $b$ determines the centre of transition, $x$ is classifier 2 output where $x = 0$ for stand-to-sit and $x = 1$ for sit-to-stand transition, gain $G$ scales the output of activation function between 0° and 90° and the thigh angle $\theta_T$ is the output.

*Estimation of the thigh angular kinematics.* Based on the classification of each segment, we used a model similar to a single neuron of an artificial neural network with appropriate activation function to estimate the thigh kinematics for each state (see Fig 4). We assumed that when a person is standing, the thigh angle, $\theta_T$ is 0° and when a person is sitting on a chair, $\theta_T$ is 90°. We confirmed this by observing the distribution of the thigh angles while sitting and standing from the reference data collected from the young healthy participants as detailed in the next section. The angular velocity $\omega_T$ and angular acceleration $\alpha_T$ were zero during the stationary states.

To estimate stand-to-sit transition angle $\theta_T$, we used an adaptable differentiable function with parameters that could be optimised to model the thigh motion kinematics. This function can be generalised and represented as an artificial neural network consisting of a single layer with a single neuron with sigmoid activation function in Eq (8) for regression (see Fig 4). This approach is generalisable and can enable more complex transitions to be modelled but is sufficient for the experiments we describe in this study. In practice, to estimate thigh angular kinematics in this study, we performed regression analysis with sigmoid model described in Eq (8). Our approach can be generalised and represented in the form of a single neuron depicted by the model in Fig 4. The model parameters $w$ and $b$ determined the speed of transition and the centre of the sigmoid curve, the midpoint of the transition segment respectively. Input to the model $t$ is the time window of transition segment. The value of $w$ indicating the speed of transition was estimated, in the range of 0 and 1, by minimising the root mean squared error between the estimated angle and the ground truth reference angle of the thigh using least squares. The optimum estimated value of $w$ was found to be 0.135. Additional model parameter $x$ is the classification result from classifier 2 where $x = 0$ for stand-to-sit and $x = 1$ for sit-to-stand transition. We chose activation function in Eq (8) to obtain a smooth transition of the angle for sit-to-stand by assuming that the angles for sit-to-stand and stand-to-sit are symmetrical for the thigh, which was also confirmed by observing the angular velocities in the ground truth reference data which were indistinguishable.

$$\theta_T = F_{SiSt}(t, x) = \frac{e^{x(-w(t-b))}}{1 + e^{-w(t-b)}} \tag{8}$$

For stand-to-sit transition, the thigh angle is modelled by Eq (9) with $x = 0$ in Eq (8).

$$\theta_T = F_{SiSt}(t, 0) = \frac{1}{1 + e^{-w(t-b)}} \tag{9}$$

For sit-to-stand transition, the thigh angle is modelled by Eq (10) with $x = 1$ in Eq (8).

$$\theta_T = F_{SiSt}(t, 1) = 1 - \frac{1}{1 + e^{-w(t-b)}} \tag{10}$$

Sigmoid function in Eq (8) is differentiable and thus $\omega_T = \frac{d\theta_T}{dt}$ and $\alpha_T = \frac{d^2\theta_T}{dt}$ for sit-to-stand and stand-to-sit are valid. This model analogous to a single layer with a single neuron was sufficient to estimate thigh angle (results are shown in the next section), however, it can be extended to a more complex artificial neural network with any bounded continuous differentiable activation function to perform regression to estimate the thigh angle.

## Ethics statement

This study was conducted in two stages with three participant groups. Ethical approval for the first stage of the study with younger healthy adults was obtained from the ethics committee of the University of Reading, UK. The ethical approval for the second stage of the study with older healthy participants and people with Parkinson's was obtained from the ethics committee of the University of Southampton, UK. Participants were provided with an information sheet detailing the purpose of the study, procedure of the experiment and nature of the data collected. Participants in all the three groups gave their informed written consent prior to their participation in the study. The individual seen in the figure in this manuscript has given written informed consent (as outlined in PLOS consent form) to publish the figure.

## Study design

**Participants.** Participants from the three groups were recruited to take part in this study which was done in two stages. In the first stage, 10 younger healthy adults (YH) (37.4 ± 9.9 years (*mean ± SD*), 4 female) participated in the study conducted at the University of Reading. All participants were over the age of 18 and in good physical health without musculoskeletal or neurological conditions.

In the second stage, 12 older healthy adults (OH) (74 ± 9.1 years, 11 female) and 12 people with Parkinson's disease (PwP) (74.3 ± 7.4 years, 6 female) participated in the study conducted at the University of Southampton. Out of the 12 PwP participants, eight participants had a score of 3 on the Hoehn and Yahr (H&Y) scale, one participant had H&Y score of 2.5, one participant had H&Y score of 2 and two participants had H&Y score of 1.5. All the participants in these two groups were over the age of 60, were able to walk independently unaided, and reported themselves to be able to perform transfers, walking and activities in standing three times over a period of approximately one hour. People with Parkinson's disease had a diagnosis made by a specialist at least 12 months prior to the study. The data from 4 OH and 10 PwP participants was collected in their home and the data from rest of the participants was collected in the laboratory.

**Equipment.** *Wearable sensors*. Wearable sensors, custom designed at the University of Reading, were used to collect the movement data in this study. Each wearable sensor consisted of a triaxial accelerometer and a triaxial gyroscope. The data were stored to the internal SD card. The sensors sampled at nominal rate of 50 Hz and ±4 g provided an actual sampling rate of 49.985 ± 0.016 Hz. The bandwidth at nominal sampling rate was 21 Hz with a noise density of 0.14 $mg/\sqrt{Hz}$. All sensor data were resampled using video recording as an external time base. Further details of the sensors can be found in [34]. The wearable sensors were attached to the shank and the back as shown in Fig 5. The individual seen in Fig 5 has given written informed consent (as outlined in PLOS consent form) to publish this figure.

(A)　　　　　　　　(B)

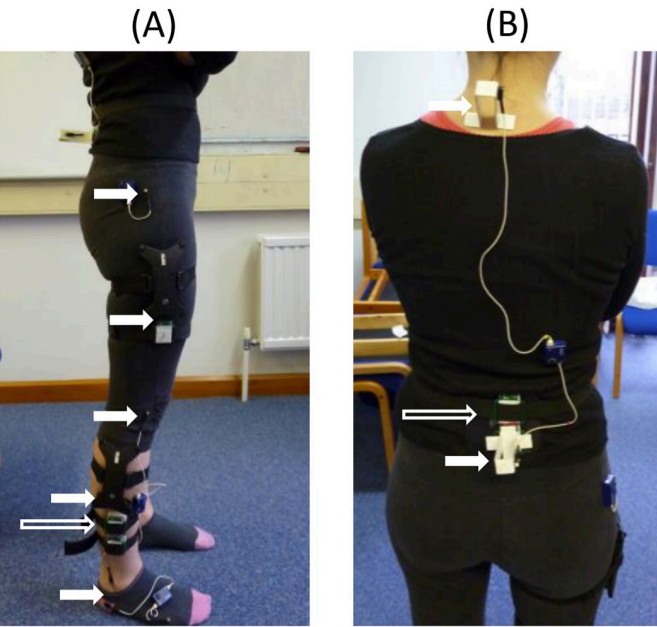

**Fig 5. Full inertial wearable sensors and Codamotion marker setup.** (A) Shank and thigh inertial sensors and Codamotion markers positions (B) Back inertial sensor and Codamotion marker positions. The long hollow arrows show the position of the inertial sensors placed on the shank and back. The solid short arrows show the position of Codamotion markers on the leg and the back.

*Motion capture*. To provide the ground truth data for validating the results of the parametric estimation models, we collected motion capture data using the Codamotion 3D Motion Analysis System (Codamotion, Rothley, UK) for the young healthy (YH) participant group. The Codamotion active markers were used to track the motions of the body during data collection. The ODIN (Codamotion, Rothley, UK) software was then used to extract the body segment angular displacement, velocity and acceleration. The Codamotion sensors were placed on the leg and back as shown in Fig 5. The motion capture data was not collected for OH and PwP groups because the data recording was done at home for several participants since these groups had difficulty travelling. Rigorously validated model for kinematics estimation using motion capture data from YH was applied to OH and PwP groups.

**Experimental protocol.** Prior to wearable sensors' data collection, the distance in meters of the inertial sensor on the shank from the ankle ($L_S$) and the distance of the inertial sensor on the back from the hip ($L_B$) were recorded for input into the modelling algorithm for each participant.

The wearable inertial sensors were attached to the right shank and to the middle of the lower back using elastic straps. The active Codamotion markers were placed on the shank and back close to the inertial sensors for YH. The clusters of Codamotion markers were also attached to the thigh where there was no inertial sensor.

The YH participants were then asked to perform three sets of five sit-to-stand and stand-to-sit transitions, with rest in between the sets as required. The OH and PwP participants were asked to perform a single set of three sit-to-stand and stand-to-sit transitions. All the inertial sensor data for YH, OH and PwP, and the Codamotion data for YH is included in the S1–S7 Files.

**Data processing.** Data obtained from the inertial sensors on the shank and the back were synchronised using ELAN software [40] by tapping the sensors together at the beginning of

the trial and at the end of the trial. The Codamotion data in the YH participants was synchronised with the inertial sensors by moving the right foot backwards and forwards at the beginning and at the end of the experiment. After aligning the data, the Codamotion data was subsampled to 50 Hz to match the sampling rate to the inertial sensors. Also, the Codamotion data was calibrated such that the angles of the shank, thigh and back were close to 0˚ in the standing position.

The measurements from the x-axis and the y-axis of the accelerometer and the z-axis of the gyroscope on the shank and the back (Fig 1) were used as measurement input to the EKF model detailed in the previous section to obtain the shank and back angular kinematics $\theta_i$, $\omega_i$ and $\alpha_i$, $i \in \{S, B\}$.

Using the angular kinematics of the shank and back, the classification was performed to obtain segments of data belonging to the standing, sitting, sit-to-stand and stand-to-sit states. For estimating thigh kinematics $\theta_T$, $\omega_T$ and $\alpha_T$, function in Eq (8) was used as detailed in the previous section to model thigh angular kinematics. Since there was no sensor on the thigh, the estimation of the kinematics was completely dependent on the estimated kinematics from the shank and the back. The estimated body kinematics were compared against reference Codamotion data in the YH participants and the model was applied to the OH and PwP participants. All the analysis was completed using MATLAB.

## Results

### Estimated angular kinematics in younger healthy adults (YH)

**Comparison of the estimated angular kinematics with the reference data in younger healthy adults (YH).** We can observe from Fig 6 that the models for the shank, thigh and back estimated the angular kinematics accurately as compared to the reference data recorded with the Codamotion system. There is some difference in the back angles as seen in Fig 6B because the Codamotion sensor shifted in the seated position when the participant's back touched the back of the chair. We can observed from Fig 6C that, the thigh kinematics were estimated accurately using the proposed integrated approach of modelling and classification despite the lack of inertial sensor on this location.

Even though the estimated kinematics of the shank and back matched the kinematics obtained from the reference Codamotion data in most cases, we observed some offset between the two in some participants as shown in Fig 6D and 6E. For estimation of the thigh angle, it was assumed that the angle is 0˚ while standing and 90˚ while sitting. However, the sitting angle depends on the posture of individual's sitting position. In Fig 6F, the individual sat in a slightly different posture with the feet tucked under the chair, making the thigh angle less than 90˚ during sitting.

Thus, the visual inspection of the plots of angular kinematics for all the trials of all the YH participants showed that the estimated kinematics matched the reference kinematics obtained from the Codamotion sensor. The next section details the quantitative results of the comparison of the estimated and reference kinematics.

**Normalised root mean squared error between estimated and reference angular kinematics.** The normalised root mean squared error (NRMSE) was calculated between the estimated angular kinematics using the model shown in Fig 2 and the reference angular kinematics obtained from motion capture for evaluating the quality of the models used for estimation. The average NRMSEs over the three runs for all the YH participants are shown in Table 1. We observed that the NRMSEs are very low with the average of 10% for angular velocity and angular acceleration. The NRMSEs for angular displacement are higher due to the offset between the inertial sensors and Codamotion markers on the shank. Thus, these NRMSE

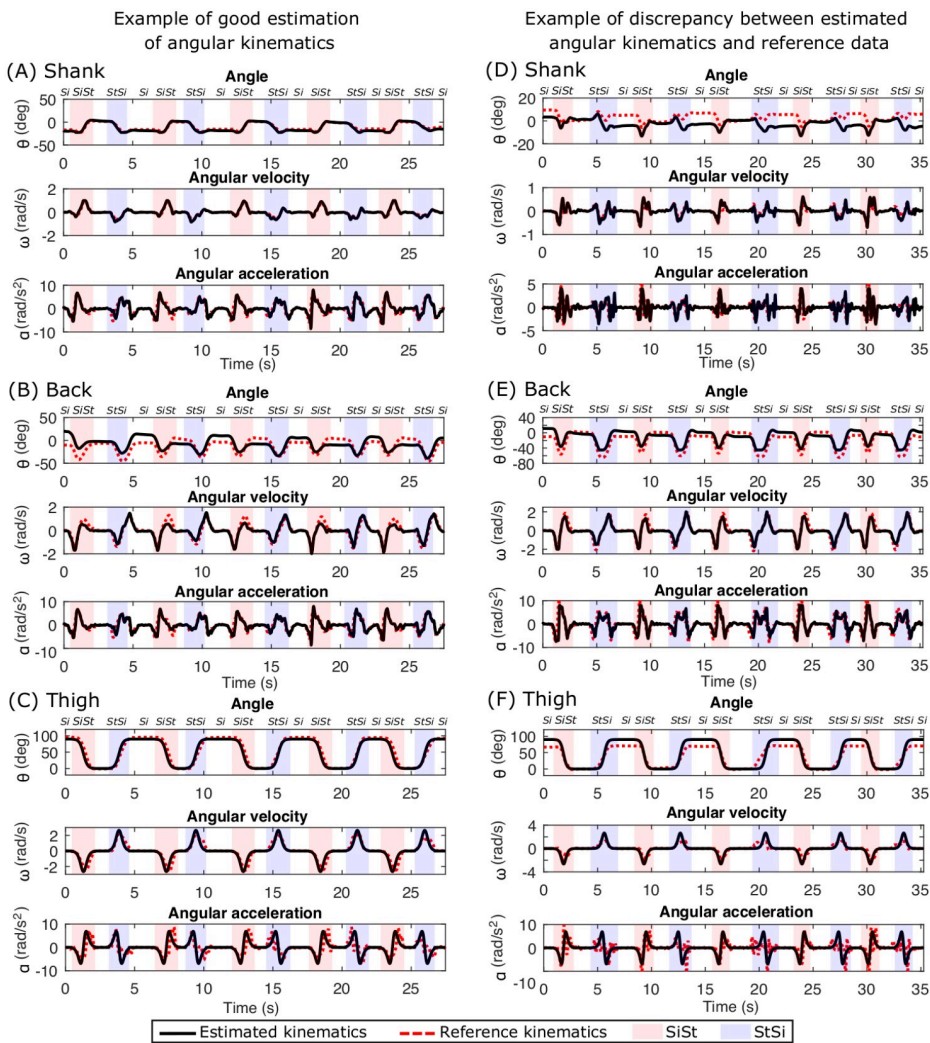

**Fig 6. Comparison of estimated kinematics with Codamotion reference kinematics in young healthy participants.**
An example of good angular kinematics estimation (left column) is shown for (A) Shank, (B) Back and (C) Thigh. The segmentation of sit-to-stand, stand-to-sit, sit and stand (no label on the top) is also shown with different background colours and labels at the top. An example of discrepancies between estimated kinematics and reference kinematics from Codamotion data (right column) is shown for (D) Shank, (E) Back and (F) Thigh.

results confirm that our proposed model estimates the angular kinematics of the three segments of the body correctly.

**Bland-Altman analysis.** The Bland-Altman plots [41] for comparing the reference angular kinematics obtained from Codamotion and the angular kinematics estimated by our proposed method in all the YH participants are shown in Fig 7. The Bland-Altman plots show that there is an agreement between the reference kinematics and estimated kinematics as the mean difference between the two is zero which is represented by the solid horizontal line and the majority of the points are located between ±2 standard deviations represented by the dotted horizontal lines. There is a small discrepancy between the reference and estimated shank since their mean difference is slightly larger than zero. The individual participants have Bland-Altman plots similar to each other and to the the Fig 7 of all the participants together.

**Table 1. Normalised root mean squared error (NRMSE) for shank, back and thigh for all the young healthy (YH) participants.**

| Participant YH | Shank NRMSE | | | Thigh NRMSE | | | Back NRMSE | | |
|---|---|---|---|---|---|---|---|---|---|
| | Angle | Angular Velocity | Angular Acc. | Angle | Angular Velocity | Angular Acc. | Angle | Angular Velocity | Angular Acc. |
| 1 | 0.1401 | 0.0465 | 0.0617 | 0.0929 | 0.1018 | 0.1745 | 0.2553 | 0.0907 | 0.0932 |
| 2 | 0.1386 | 0.0615 | 0.0520 | 0.1373 | 0.0877 | 0.1012 | 0.1705 | 0.0549 | 0.0942 |
| 3 | 0.3683 | 0.0821 | 0.1366 | 0.0953 | 0.0914 | 0.1202 | 0.1197 | 0.0585 | 0.1385 |
| 4 | 0.3537 | 0.0585 | 0.1348 | 0.0718 | 0.0722 | 0.1239 | 0.1075 | 0.0542 | 0.1049 |
| 5 | 0.3464 | 0.1277 | 0.1597 | 0.1013 | 0.0745 | 0.0917 | 0.2090 | 0.0962 | 0.1750 |
| 6 | 0.2963 | 0.0848 | 0.0446 | 0.2457 | 0.1310 | 0.1409 | 0.1549 | 0.0760 | 0.0696 |
| 7 | 0.3163 | 0.0746 | 0.1113 | 0.1157 | 0.1304 | 0.1928 | 0.1348 | 0.0513 | 0.0902 |
| 8 | 0.2363 | 0.0439 | 0.0940 | 0.1156 | 0.0963 | 0.1272 | 0.1160 | 0.0624 | 0.1376 |
| 9 | 0.3191 | 0.0559 | 0.1229 | 0.1504 | 0.1299 | 0.1477 | 0.1476 | 0.0694 | 0.1501 |
| 10 | 0.4788 | 0.0830 | 0.0916 | 0.1906 | 0.1160 | 0.1245 | 0.2009 | 0.0621 | 0.0948 |
| **Average** | **0.2994** | **0.0718** | **0.1009** | **0.1317** | **0.1032** | **0.1345** | **0.1616** | **0.0676** | **0.1148** |
| **Std. Dev.** | 0.1004 | 0.0247 | 0.0390 | 0.0524 | 0.0226 | 0.0309 | 0.0476 | 0.0155 | 0.0333 |

Thus using NRMSE and Bland-Altman analysis, we validated the wearable inertial sensors against the reference data from young individuals, and our proposed angular kinematics estimation model was found to be stable in this context.

## Estimated angular kinematics in older healthy adults (OH) and people with Parkinson's (PwP)

We applied the integrated modelling and classification method on the OH and PwP participant groups to estimate their angular kinematics during sit-to-stand transitions. An example of the estimated angular kinematics for OH and PwP is given in Fig 8A–8C and 8D–8F respectively. The proposed method was able to reliably classify the sit-to-stand and stand-to-sit transitions in both OH and PwP groups and thus estimate their angular kinematics. We observed that the angular kinematics in these two groups were not as smooth as those in the YH group. The fluctuations were visually observed especially in the angular velocity and angular acceleration Fig 8A, 8B, 8D and 8E which could be an indication of an overall instability during the sit-to-stand movements or tremors in the PwP group. In many participants, this instability was also observed in the stationary states. The PwP group showed more instability than the OH group upon visual inspection as seen in Fig 8D and 8E. We observed that in many participants from these two groups, there was a brief pause during the sit-to-stand and stand-to-sit transitions which may indicate that the older adults perform these transitional movements more statically by keeping their accelerations low and making their velocity zero half way through the transition. This is an interesting finding giving an insight into the strategies used by different groups for performing sit-to-stand transitions and will require further investigation.

## Comparison of results in younger healthy adults (YH), older healthy adults (OH) and people with Parkinson's (PwP)

**Classification accuracies.** The two-tiered classification approach successfully segmented and classified sit-to-stand motion in sitting, standing, sit-to-stand and stand-to-sit stages in all the three participant groups with high accuracy. The best classification accuracies were obtained in the YH group. The classification accuracies together for all the four states are 98.67%, 94.20% and 91.41% for YH, OH and PwP respectively. There were no false positives in stationary and transition state classifications. The misclassifications occurred when the sitting

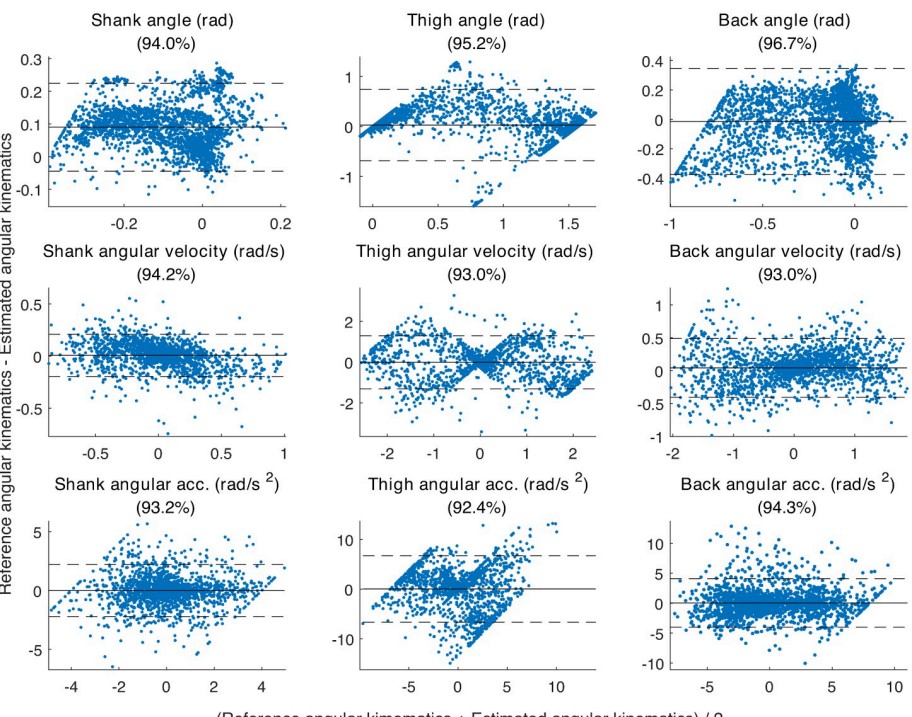

**Fig 7. Bland-Altman plots.** The Bland-Altman plots showing comparison between the reference Codamotion angular kinematics and estimated angular kinematics using the proposed integrated approach of modelling and classification for the shank, thigh and back. The x-axis shows the mean of the two measures and the y-axis shows the difference between the two measures. The solid horizontal represents the mean difference between the reference and estimated kinematics and the dotted horizontal lines show the ±2 standard deviation boundaries. A Bland-Altman plot typically looks for points to be within ±2 standard deviations of the mean difference, the title of each sub-figure has this as percentage.

and standing postures were identical and indistinguishable. This happened when participants sat upright. In such cases, the sensor data was also identical for the sit-to-stand transitions.

**Timings for sit-to-stand and stand-to-sit transitions.** The average time taken for sit-to-stand was $1.44 \pm 0.36$ s, $1.80 \pm 0.54$ s and $2.29 \pm 1.44$ s in YH, OH and PwP respectively. The average time taken for stand-to-sit was $1.55 \pm 0.33$ s, $1.77 \pm 0.65$ s and $2.18 \pm 0.84$ s for YH, OH and PwP respectively. The YH group took the least amount of time to perform transitions and approximately equal amount of time for sit-to-stand and stand-to-sit. The OH group performed the transitions slower than the YH group and took approximately same time for both the transitions. The PwP group took more time to perform sit-to-stand transitions than the rest of the two groups. The timings for the YH for different participants is consistent with a small standard deviation as compared to the other groups. The variability in the timings to perform the transitions gradually increases from YH, OH to PwP group. This is shown in the box plot in the Fig 9. Comparing the timings of three groups during sit-to-stand and stand-to-sit independently using Mann-Whitney U test and Bonferroni correction for multiple comparisons between the three groups revealed that sit-to-stand timings of YH and PwP were significantly different ($p < 0.001$) and OH and PwP were also significantly different ($p < 0.05$). Also, during stand-to-sit, timings of YH and OH were significantly different ($p < 0.001$) and timings of YH and OH were significantly different ($p < 0.001$).

**Variability in the posture of sitting and standing.** The box plots in Fig 10A and 10B show the posture variability in the YH, OH and PwP groups during sitting and standing

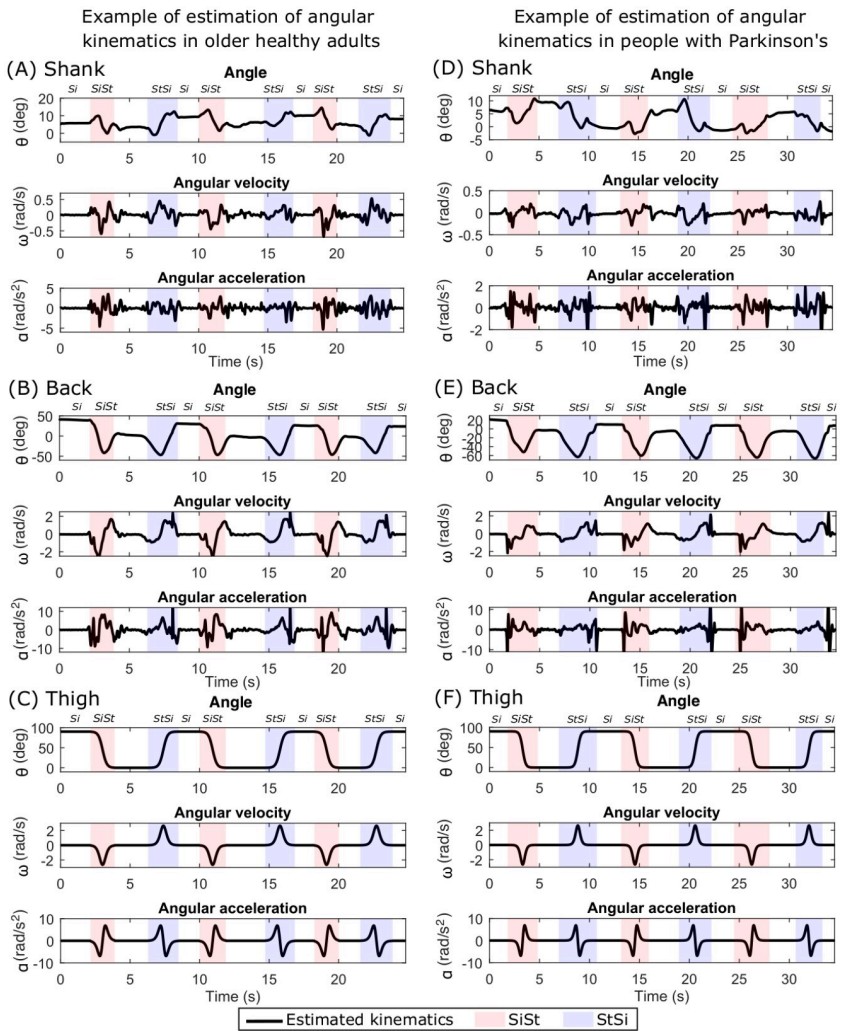

**Fig 8. Estimated angular kinematics in older healthy adults (OH) and people with Parkinson's (PwP).** An example of estimated angular kinematics in OH participants (left column) is shown for (A) Shank, (B) Back and (C) Thigh. The segmentation of sit-to-stand, stand-to-sit, sit and stand (no label on the top) is also shown with different background colours and labels at the top. An example of estimated angular kinematics in PwP participants (right column) is shown for (D) Shank, (E) Back, and (F) Thigh.

respectively. The variability in the average angles of the shank and the back differ in three groups. Overall, the back angles show higher variability. The variability is higher in the OH and PwP groups suggesting that the older adults have variable postures during sitting and standing depending on their physical condition and due to change in their centre of gravity in order to maintain their balance. The sitting position shows more postural variability (see Fig 10A) because of the differences in the sitting styles such as hunching, leaning back on the chair, tucking their feet under the chair or sitting very upright. However, this variability is not present in the YH group in the standing posture in contrast to OH and PwP groups during standing. The Mann-Whitney U test with Bonferroni correction for multiple comparisons showed that the shank and back angles during sitting and standing were significantly different between YH and PwP ($p < 0.01$) and also between the other groups in some cases as shown in Fig 10. This shows that the age and the physical condition affects the posture which can be detected by the estimated angular kinematics.

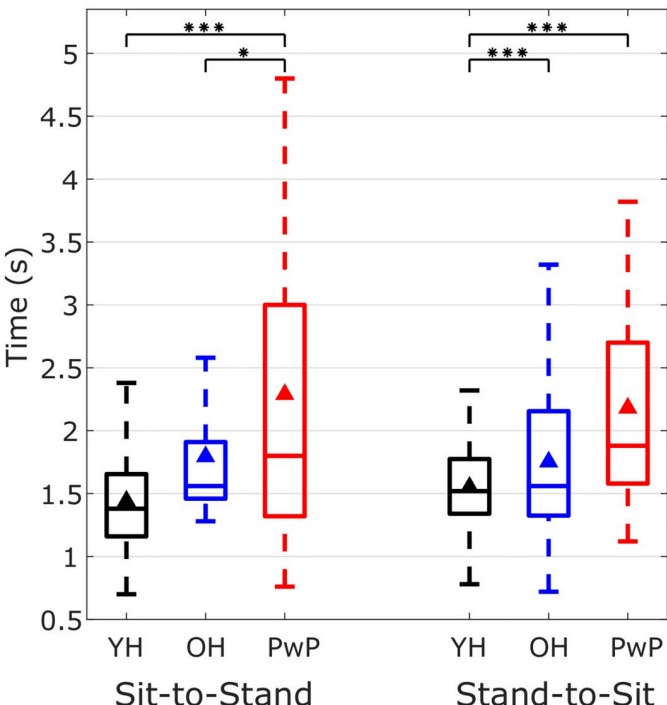

**Fig 9. Timings of sit-to-stand and stand-to-sit in younger healthy (YH) adults, older healthy (OH) adults and people with Parkinsons (PwP).** The time taken by participants to perform sit-to-stand and stand-to-sit from all the three YH, OH and PwP groups. The black triangle shows the mean time. Statistically significant differences (Mann-Whitney U test with Bonferroni correction for multiple tests) in the timings among the three groups for sit-to-stand and stand-to-sit are shown by the stars indicting the p-values (one star indicates $p < 0.5$, two stars indicate $p < 0.01$ and three stars indicate $p < 0.001$).

**Differences in sit-to-stand and stand-to-sit transitions in younger healthy adults (YH), older healthy adults (OH) and people with Parkinson's (PwP).** The Fig 11A–11H show the grand average sit-to-stand and stand-to-sit angular velocities and angles respectively for the shank and the back. The velocities are highest in the YH and lowest in PwP for the shank and the back during both sit-to-stand and stand-to-sit (see Fig 11A–11D). This suggests that the OH and PwP perform statically stable transitions in order to maintain their balance. YH have greater angel for the shank during sit-to-stand transitions than the other groups (see Fig 11E and 11F). The shank angles are very small in the PwP group (Fig 11E–11H). Thus, angular kinematics can inform us about the differences in sit-to-stand transitions in the three groups.

## Discussion

Expanding on our previous work to estimate two-segment upper limb kinematics using one inertial sensor on each limb segment [34], in this study, we have developed a three-segment body model integrated with classifier to estimate the angular kinematics and classify sit-to-stand motion using only two inertial sensors placed on the shank and back. This provides a low-cost solution to model the sit-to-stand activities which are crucial for human mobility and are often affected due to old age and motor impairment. This approach combines both kinematic analysis and movement classification approaches, and thus could be employed for monitoring the quality of movement as well as assessing general movement patterns and may therefore be of value in clinical decision making. We have demonstrated that our approach of combining the model and the classification for estimation of kinematics is robust and stable

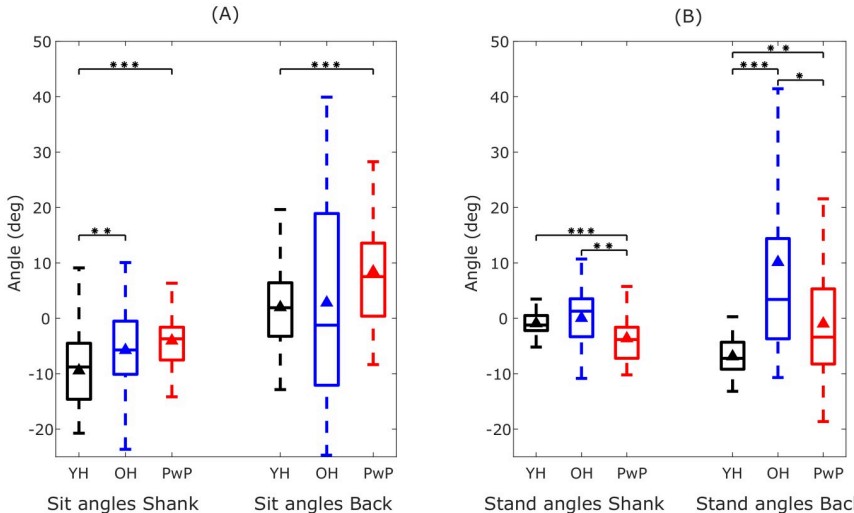

**Fig 10. Variability in the posture during sitting and standing in the shank and back in younger healthy (YH) adults, older healthy (OH) adults and people with Parkinson's (PwP).** (A) The average angles of the shank and back in YH, OH and PwP participants during sitting. The black triangle shows the mean angle. (B) The average angles of the shank and back in YH, OH and PwP participants during standing. Statistically significant differences (Mann-Whitney U test with Bonferroni correction for multiple tests) in the angles of the shank and back among the three groups during sitting and standing are shown by the stars indicting the p-values (one star indicates $p < 0.5$, two stars indicate $p < 0.01$ and three stars indicate $p < 0.001$).

from its successful application across all the three participant groups comprising of younger healthy adults, older healthy adults and people with Parkinson's. Our approach can be generalised across populations for modelling sit-to-stand kinematics.

We have chosen a simplified 2-dimensional model because the sit-to-stand transitions occur predominantly in the sagittal plane [22]. The sagittal movement assumption is common for estimating kinematics since finding 3-dimensional kinematics poses difficulty when using body fixed inertial sensors [2, 42]. Further work is needed to include out of plane information, however, adding an extra dimension might not give more information about sit-to-stand motion, although may be helpful when considering combined sit to stand and turning movements. The 3-dimensional model is likely to be of particular relevance when considering movement dynamics in the younger healthy participants. However, we are developing this model to study the OH and the PwP groups that are less dynamic and hence their sit-to-stand transitions are restricted to the sagittal plane for which our 2-dimensional model is sufficient.

The EKF based model successfully estimated the kinematics of the shank and back which was confirmed by comparing the outcome to the Codamotion reference data as shown in Fig 6A and 6B. The EKF models the measurement noise and the process noise to estimate accurate results, unlike the approach of obtaining kinematics directly from accelerometer measurements which needs explicit de-noising due to the noise and drifts in the sensors, and differentiation and integration of their output. Our model is insensitive to the distances of sensors $L_i$, $i \in \{S, B\}$ from the ankle and hip (see Fig 1). This shows that the model is robust across people with different anatomical measurements and the location of sensor placement on the body segment.

In our model, we have ignored the translation and acceleration of the hip during sit-to-stand transitions. This has introduced bias in the data. However, by comparing the results of YH participants with the reference Codamotion data (Figs 6 and 7 and Table 1), we observe that the bias is insignificant enough to allow this assumption. The bias will be higher when the

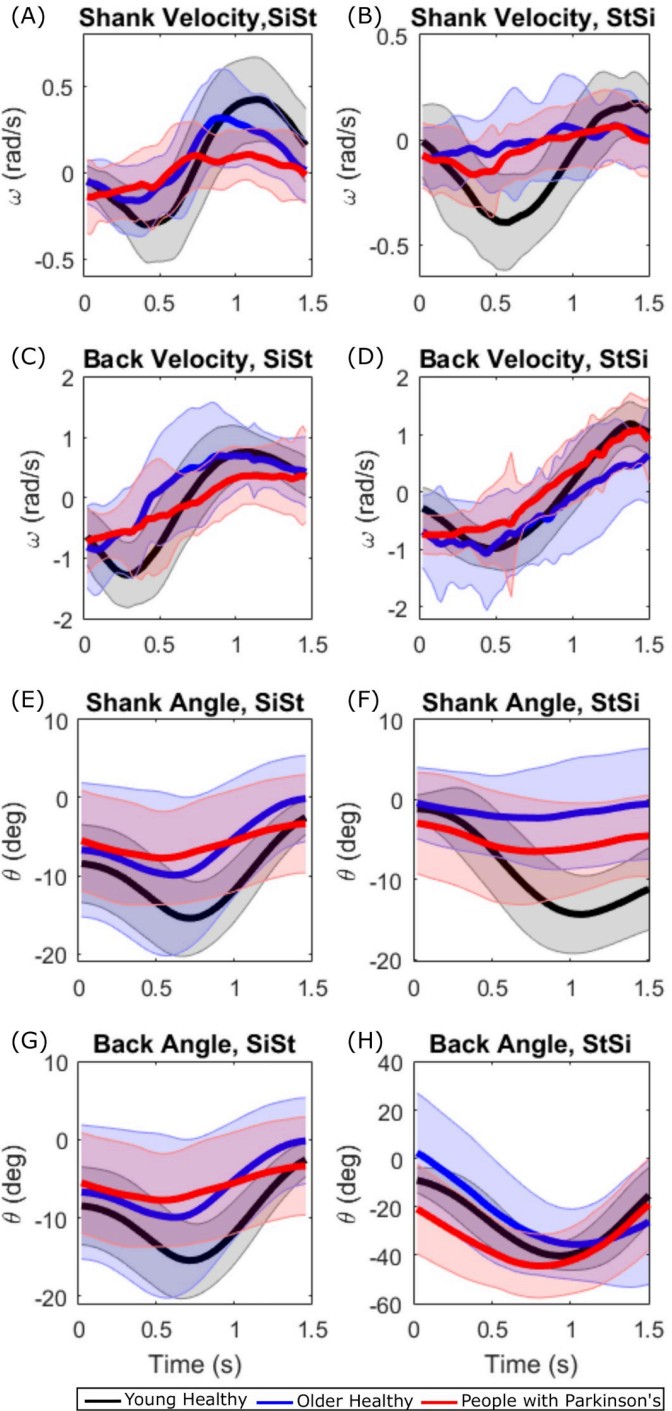

**Fig 11. Grand average shank and back velocity and angles during sit-to-stand and stand-to-sit in younger healthy (YH) adults, older healthy (OH) adults and people with Parkinson's (PwP).** (A) Grand average shank velocities in the three participant groups during sit-to-stand. (B) Shank velocities during stand-to-sit. (C) Back velocities during sit-to-stand. (D) Back velocities during stand-to-sit. (E) Shank angles during sit-to-stand. (F) Shank angles during stand-to-sit. (G) Back angles during sit-to-stand. (H) Back angles during stand-to-sit.

accelerations are high, consequently since the OH and particularly PwP groups have lower accelerations, the bias will also be low in these groups and can be disregarded.

The model was validated on the YH participants and then applied on the OH and PwP participants. We were not able to do 3-dimensional motion capture with the Codamotion system in OH and PwP groups, primarily because the data was collected from people's homes. This was seen as appropriate since the participant groups had difficulty travelling. We aimed to collect the data with minimal disruption and given the problems of setting up the Codamotion system in the home environment, we intentionally omitted this stream of data. However, the validation of wearable sensors using motion capture data collected from YH group gave a strong indication that our kinematics estimation method using the proposed model works as seen from the small NRMSE (Table 1) and Bland-Altman plots (Fig 7) and hence further validation for the OH and PwP groups was not required.

We minimised the number of sensors required to estimate the kinematics of three-segment model. We chose to place sensors on shank and back because only this combination can model three-segment kinematics by estimating thigh kinematics. If two consecutive segments were chosen, the third could not be detected. It was also easier to place sensors on the shank and the back than thigh, because of larger muscle movements in thigh during sit-to-stand and discomfort during sitting with a sensor on thigh. We have dealt with a difficult problem of estimating thigh kinematics effectively without placing an inertial sensor on this location. Estimating thigh kinematics from this missing data is challenging because it is an ill-posed problem and the thigh angle has infinitely many solutions in the range between 0˚ and 90˚ for sit-to-stand activity. To deal with this, we have incorporated a classification based approach where we identify the current state (sit, stand, sit-to-stand or stand-to-sit) and apply different models to the individual state. Thus, we uniquely combine two challenges: classification of different stages in sit-to-stand movement and obtaining angular kinematics for the three-segment body model.

Even though the estimated thigh kinematics are accurate with small average error of 13% for the angle, angular velocity and angular acceleration of the shank, thigh and back in comparison to the reference motion capture data in YH Table 1), we have based it on the observational assumptions that when the person is seated on the chair, the angle of the thigh is 90˚ and when the person is standing, the thigh angle is 0˚. This highly depends on the posture of an individual while sitting or standing which is observed in Fig 6F where the sitting angle for thigh is slightly less than 90˚. Since, sitting and standing postures differ from person to person, this approach might not yield accurate results in the cases for postural defects, specifically in the OH and PwP groups. We have modelled the transitions between 0˚ and 90˚ for sit-to-stand and stand-to-sit by using a single neural network node with a sigmoid activation function in Eq (8) with an input from the sit-to-stand transition classifier. This sigmoid function was chosen because it is continuous and differentiable, leading to smooth transitions between sitting and standing states. The assumption of symmetrical sit-to-stand and stand-to-sit transitions allowed us to estimate the parameter $w$ only once. This assumption of symmetry might not be true for OH and PwP. This model can be generalised and extended with a more complex artificial neural network for regression with any other suitable continuously differentiable activation functions.

We have achieved high sit-to-stand classification accuracies of 98.67%, 94.20% and 91.41% for YH, OH and PwP respectively using a two-tiered classification with k-means clustering. This unsupervised learning approach allowed us to classify movements with high accuracy on individual participants with a large inter-participant variability using as few as two repetitions of movements. This could be useful in clinical settings where collecting large amount of movement data and training machine learning models is not feasible due to time limitations.

The OH and PwP groups showed higher variability in the average time taken to perform sit-to-stand transitions (Fig 9) and also in the angles of the shank and especially back while sitting and standing (Fig 10). These variabilities can be attributed to the differences in the mobility levels in the OH group and the varying effect of Parkinson's disease on mobility in the PwP group which may also affect their posture. The YH group has a consistent posture during standing while the OH and PwP groups show larger variability in their average back angle (Fig 10B) because of the instability during standing. The PwP group showed very low accelerations and low velocities (see Fig 11) leading to performing statically stable movements by pausing in the middle of sit-to-stand and stand-to-sit transitions. Thus, the angular kinematics of the three-segment body model in the sagittal plane provide insights into differences in sit-to-stand transitions in different groups varying in age and functional mobility.

The ability to stand up from sitting indicates balance control and functional lower limb strength. The inability to stand up from sitting and display of unsteadiness when completing the task suggests the person is more likely to have restricted mobility and could be at risk of falling. Hence it is important to be able to assess sit-to-stand performance as it can allow the identification of persons at risk. Our proposed method allows detailed assessment of sit-to-stand motion by modelling all the three segments of the body involved in this motion whilst minimising the number of sensors needed for sit-to-stand tests. Our proposed method can hence be used as a tool alongside other methodologies for assessment of sit-to-stand transitions. Our novel combined approach facilitates comprehensive study of sit-to-stand movement in varying demographics of people which not only models movements providing their continuous kinematics, but also segments and classifies individual movements and computes time taken for individual transfers. Our proposed method is not restricted to sit-to-stand movement and can also be extended to have broader applications in sports science to study a range of different motions. Finally, our proposed method will allow long term (multi-day to multi-week) movement studies to be conducted since these sensors are unobtrusive and easy to wear.

## Conclusion

In this paper, we have proposed a novel integrated approach for estimating the body kinematics during sit-to-stand transition motions using only two wearable inertial sensors with a triaxial accelerometer and a triaxial gyroscope each. This provides an inexpensive and portable way of estimating human motion as opposed to expensive optic motion sensor systems requiring a complex setup or placing a sensor on each segment of the body. The two wearable sensors are comfortable for prolonged use and require low power to operate.

A robust three-segment body kinematic model is formed based on limb kinematic model and parameter estimation using EKF. We have tested this model on the three groups of young healthy adults, older healthy adults and people with Parkinson's disease. We have solved two challenges of modelling and classification of sit-to-stand and stand-to-sit movements by incorporating classifier in the estimation of the body kinematics. Our model not only estimates the kinematics on the shank and the the back accurately, but also the kinematics for thigh which is an ill-posed problem as there is no inertial sensor on this location. Thus, our model effectively deals with the missing data, and at the same time, segments and classifies the sit-to-stand and stand-to-sit, standing and sitting states robustly using unsupervised learning with an accuracy of 98.67%, 94.20% and 91.41% for YH, OH and PwP respectively. The estimated kinematics are similar to the ground truth kinematics obtained from the commercial Codamotion system as compared in YH participants.

## Supporting information

**S1 File. Young Healthy (YH) shank inertial sensors data.**
(XLSX)

**S2 File. Young Healthy (YH) back inertial sensors data.**
(XLSX)

**S3 File. Older Healthy (OH) shank inertial sensor data.**
(XLSX)

**S4 File. Older Healthy (OH) back inertial sensor data.**
(XLSX)

**S5 File. People with Parkinson's (PwP) shank inertial sensordata.**
(XLSX)

**S6 File. People with Parkinson's (PwP) back inertial sensordata.**
(XLSX)

**S7 File. Young Healthy (YH) reference Codamotion angular kinematics.**
(XLSX)

**S1 Text.**
(TXT)

## Acknowledgments

The authors would like to thank all the participants in this study who helped in recording the data and enabled this research.

## Author Contributions

**Conceptualization:** Maitreyee Wairagkar, Emma Villeneuve, Rachel King, Balazs Janko, Veena Agarwal, William Holderbaum, William S. Harwin.

**Data curation:** Emma Villeneuve, Rachel King, Balazs Janko, Malcolm Burnett, Veena Agarwal, Dorit Kunkel.

**Formal analysis:** Maitreyee Wairagkar.

**Funding acquisition:** Ann Ashburn, William Holderbaum, William S. Harwin.

**Investigation:** Maitreyee Wairagkar, Emma Villeneuve, Rachel King, Malcolm Burnett, Veena Agarwal.

**Methodology:** Maitreyee Wairagkar, Emma Villeneuve, Rachel King, William Holderbaum, William S. Harwin.

**Project administration:** Malcolm Burnett, Dorit Kunkel.

**Resources:** Balazs Janko, Malcolm Burnett.

**Software:** Maitreyee Wairagkar, Rachel King.

**Supervision:** Dorit Kunkel, Ann Ashburn, R. Simon Sherratt, William Holderbaum, William S. Harwin.

**Validation:** Maitreyee Wairagkar, Balazs Janko, Dorit Kunkel, Ann Ashburn, R. Simon Sherratt, William Holderbaum, William S. Harwin.

**Visualization:** Maitreyee Wairagkar.

**Writing – original draft:** Maitreyee Wairagkar.

**Writing – review & editing:** Emma Villeneuve, Rachel King, Veena Agarwal, Dorit Kunkel, Ann Ashburn, R. Simon Sherratt, William Holderbaum, William S. Harwin.

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
