## [Decision Letter · Decision Letter 0]

25 Oct 2021

PONE-D-21-23075A novel approach for modelling and classifying sit-to-stand kinematics using inertial sensorsPLOS ONE

Dear Dr. Wairagkar,

Thank you for submitting your manuscript to PLOS ONE. After careful consideration, we feel that it has merit but does not fully meet PLOS ONE’s publication criteria as it currently stands. Therefore, we invite you to submit a revised version of the manuscript that addresses the points raised during the review process.

We look forward to receiving your revised manuscript.

Kind regards,

Anwar P.P. Abdul Majeed

Academic Editor

PLOS ONE

2. We note that Figure 5 includes an image of a [patient / participant / in the study].

Reviewers' comments:

Reviewer's Responses to Questions

**Comments to the Author**

1. Is the manuscript technically sound, and do the data support the conclusions?

Reviewer #1: Yes

Reviewer #2: Yes

2. Has the statistical analysis been performed appropriately and rigorously? 

Reviewer #1: Yes

Reviewer #2: Yes

3. Have the authors made all data underlying the findings in their manuscript fully available?

Reviewer #1: Yes

Reviewer #2: Yes

4. Is the manuscript presented in an intelligible fashion and written in standard English?

Reviewer #1: Yes

Reviewer #2: Yes

5. Review Comments to the Author

Reviewer #1: In line 240, the authors mention they employed a single neuron ANN model, please clarify if a single hidden layer with single neuron is used. if it is so, please explicitly mention it in the manuscript.

What is the learning algorithm used for the ANN model.

Reviewer #2: The authors explores the employment of an unsupervised machine learning model in classifying sit-to-stand movements based on kinematics data captured and reconstructed using reduced inertial sensors. The paper is well written, the methodology employed is sound and the results were well discussed. The following are some minor comment(s):

1. Line 192 - what linear classifier was used here?

2. Line 257 - Was a single hidden layer with one hidden neuron used to estimate the thigh angle? You did mention that it was deemed sufficient, how was this conclusion made? Also do report what optimisation algorithm used, e.g. Quasi-Newton, LM?

6. PLOS authors have the option to publish the peer review history of their article (what does this mean?). If published, this will include your full peer review and any attached files.

Reviewer #1: No

Reviewer #2: No

---

## [Author Response · Author response to Decision Letter 0]

22 Dec 2021

Dear Editor and Reviewers,

We would like to thank the reviewers and editor for their comments and feedback on our paper “A novel approach for modelling and classifying sit-to-stand kinematics using inertial sensors” submitted to PLOS ONE. We accept and are grateful for all comments, which have resulted in a stronger revised manuscript. 

A detailed response to all reviewer comments, organised and numbered by reviewer follows. Comments from reviewers are quoted directly in bold, followed with a complete response to each critique. For the convenience of reviewers, we have also quoted new text from manuscript directly relevant to each comment in italics as a part of our responses. All revisions are highlighted in the manuscript text in red. 

Reviewer #1 (Comments to authors):

1) In line 240, the authors mention they employed a single neuron ANN model, please clarify if a single hidden layer with single neuron is used. if it is so, please explicitly mention it in the manuscript.

-- We thank the reviewer for their constructive comments and feedback. We have now clarified that the representation of our model could be generalised to artificial neural network with a single layer with a single neuron with sigmoid activation for regression to estimate thigh movement angular kinematics as follows in the manuscript in line 238 as follows:

“To estimate stand-to-sit transition angle �T, we used an adaptable differentiable function with parameters that could be optimised to model the thigh motion kinematics. This function can be generalised and represented as an artificial neural network consisting of a single layer with a single neuron with sigmoid activation function in Eq 8 for regression (see Fig 4).”

2) What is the learning algorithm used for the ANN model.

-- We have estimated the angular kinematics of thigh using regression to fit sigmoid curve. The parameters of sigmoid model were estimated by least squares. We have now added the following description in line 244:

“In practice, to estimate thigh angular kinematics in this study, we performed regression analysis with sigmoid model described in Eq (8). Our approach can be generalised and represented in the form of a single neuron depicted by the model in Fig 4. The model parameters w and b determined the speed of transition and the centre of the sigmoid curve, the midpoint of the transition segment respectively. Input to the model t is the time window of transition segment. The value of w indicating the speed of transition was estimated, in the range of 0 and 1, by minimising the root mean squared error between the estimated angle and the ground truth reference angle of the thigh using least squares.”

Reviewer #2 (Comments to authors):

The authors explore the employment of an unsupervised machine learning model in classifying sit-to-stand movements based on kinematics data captured and reconstructed using reduced inertial sensors. The paper is well written, the methodology employed is sound and the results were well discussed. The following are some minor comment(s):

-- We thank the reviewer for their encouraging feedback. We have addressed all comments as follows.

1) Line 192 - what linear classifier was used here?

-- We simply classified the stationary and transition states based on automatically identified threshold. We have now clarified it in the manuscript in line 191 as follows:

“A threshold-based binary linear classification was performed to identify stationary state for feature values below the threshold and transition state for feature values above the threshold.”

2) Line 257 - Was a single hidden layer with one hidden neuron used to estimate the thigh angle? You did mention that it was deemed sufficient, how was this conclusion made? 

-- Yes, our model can be represented as a single layer ANN with only one neuron to estimate the thigh angle. We have now clarified it in lines 238 and 262 as follows. This model was deemed sufficient for estimating simple sit-to-stand and stand-to-sit transitions based on the results described in Fig 6 comparing ground truth kinematics with estimated kinematics, Fig 7 Bland-Altman plots, and Table 1 root mean squared errors which show that the estimated output was accurate as compared to the ground truth motion capture data and the error between the two was low.

Line 238: “To estimate stand-to-sit transition angle �T, we used an adaptable differentiable function with parameters that could be optimised to model the thigh motion kinematics. This function can be generalised and represented as an artificial neural network consisting of a single layer with a single neuron with sigmoid activation function in Eq 8 for regression (see Fig 4).”

Line 262: “This model analogous to a single layer with a single neuron was sufficient to estimate thigh angle (results are shown in the next section), however, it can be extended to a more complex artificial neural network with any bounded continuous differentiable activation function to perform regression to estimate the thigh angle.”

3) Also do report what optimisation algorithm used, e.g., Quasi-Newton, LM?

-- We have used least squares for regression to fit sigmoid curve which was used to estimate the angular kinematics of thigh. We have now added the following description to clarify this in line 244:

“In practice, to estimate thigh angular kinematics in this study, we performed regression analysis with sigmoid model described in Eq (8). Our approach can be generalised and represented in the form of a single neuron depicted by the model in Fig 4. The model parameters w and b determined the speed of transition and the centre of the sigmoid curve, the midpoint of the transition segment respectively. Input to the model t is the time window of transition segment. The value of w indicating the speed of transition was estimated, in the range of 0 and 1, by minimising the root mean squared error between the estimated angle and the ground truth reference angle of the thigh using least squares.”

We have addressed all reviewer comments and revised our manuscript based on the reviewers’ feedback and journal style requirements. We thank the reviewers and the editor again and hope the revised manuscript meets their expectations. 

Thank you, 

Maitreyee Wairagkar, PhD 

Department of Mechanical Engineering

Imperial College London, 

London SW7 1AL,

UK

---

## [Editor Report · Decision Letter 1]

4 Feb 2022

A novel approach for modelling and classifying sit-to-stand kinematics using inertial sensors

PONE-D-21-23075R1

Dear Dr. Wairagkar,

We’re pleased to inform you that your manuscript has been judged scientifically suitable for publication and will be formally accepted for publication once it meets all outstanding technical requirements.

Kind regards,

Anwar P.P. Abdul Majeed

Academic Editor

PLOS ONE

Additional Editor Comments (optional): Nil

Reviewers' comments:

The author(s) have addressed the queries raised by the reviewers.

---

## [Editor Report · Acceptance letter]

15 Mar 2022

PONE-D-21-23075R1 

A novel approach for modelling and classifying sit-to-stand kinematics using inertial sensors 

Dear Dr. Wairagkar:

I'm pleased to inform you that your manuscript has been deemed suitable for publication in PLOS ONE. Congratulations! Your manuscript is now with our production department. 

Kind regards, 

on behalf of

Dr. Anwar P.P. Abdul Majeed 

Academic Editor

PLOS ONE